# Adherence to behavioral Covid-19 mitigation measures strongly predicts mortality

**Jürgen Margraf**[1]*, **Julia Brailovskaia**[1], **Silvia Schneider**[2]

**1** Department of Clinical Psychology and Psychotherapy, Mental Health Research and Treatment Center, Ruhr-Universität Bochum, Bochum, Germany, **2** Department of Clinical Child and Adolescent Psychology, Mental Health Research and Treatment Center, Ruhr-Universität Bochum, Bochum, Germany

* Juergen.Margraf@rub.de

## Abstract

In the absence of vaccines or causal therapies, behavioral measures such as wearing face masks and maintaining social distance are central to fighting Covid-19. Yet, their benefits are often questioned by the population and the level of adherence to the measures is variable. We examined in representative samples across eight countries ($N = 7,568$) whether adherence reported around June 1, 2020 predicted the increase in Covid-19 mortality by August 31, 2020. Mortality increased 81.3% in low adherence countries (United States, Sweden, Poland, Russia), 8.4% in high adherence countries (Germany, France, Spain, United Kingdom). Across countries adherence and subsequent mortality increases correlated with $r = -0.91$. No African or South American countries were included in the present study, which limits the generalizability of the findings. While reported Covid-19 mortality is likely to be influenced by other factors, the almost tenfold difference in additional mortality is significant, and may inform decisions when choosing whether to prioritize individual liberty rights or health-protective measures.

**Data Availability Statement:** All relevant data are within the manuscript and its Supporting information files.

**Funding:** The author received no specific funding for this work.

## Introduction

Given the lack of vaccines or specific causal therapies, many governments and scientists consider behavioral measures such as wearing face masks, maintaining distance from other people, avoiding large social gatherings and practicing increased hygiene to be the key tools in the fight against the Covid-19 pandemic [1]. Such "nonpharmaceutical interventions (NPIs)" are designed to slow the spread of Covid-19 by reducing physical contact within the population and by reducing uptake of the virus via droplet infection and inhalation [1, 2].

Early experiences from China pointed to the effectiveness of massive NPIs in shortening the serial interval of SARS-CoV-2 infections over time and in reducing the transmission of the virus [3–5]. Beginning in March 2020, national lockdowns were declared by many governments outside of China [6]. The exact extent and timing of measures to reduce the spread of Covid-19 varied between and even within countries [2, 7]. Since May 2020, some countries have started to ease or lift some of the government implemented anti-pandemic measures. Other measures such as keeping people at a distance and wearing face masks on public

**Competing interests:** The authors have declared that no competing interests exist.

transport, in stores or even in all public places, have been maintained or reintroduced in view of the renewed rise in infection rates. However, more or less mandatory NPIs represent restrictions on freedom and, in their strong form, have massive economic impacts [8–10]. It is therefore not surprising that they are often controversial, and are not universally adopted by all governments and citizens [2, 10–12]. For example, many Western governments and health authorities, as well as the World Health Organization, initially made contradictory or ambivalent statements about the wearing of face masks, which led to misunderstandings and even stigmatization [2]. Indeed, the justification of invasive NPIs depends on their promise of sufficient benefit, and this is precisely what critics have repeatedly questioned. While NPIs can be useful in theory and their probable effect can be modelled mathematically, strong empirical evidence is scarce [1, 13, 14]. Randomized controlled trials (RCTs) would provide particularly meaningful evidence, but in the current explosive pandemic RCTs are encountering practical and ethical challenges that could be responsible for their absence to date [1]. Quasi-experimental designs ("natural experiments") are therefore of great importance. One such study has recently shown a stronger decline in daily Covid-19 growth rates in 15 U.S. states following the introduction of face masks in public places compared to states that did not require this [11]. In addition, a strong negative correlation between the number of Covid-19 cases and lockdown measures was observed across 49 countries [7]. Model simulations specifically of wearing face masks [13] show that the community-wide benefit is likely to be greatest when face masks are used in conjunction with other NPIs, when adoption is nearly universal (nation-wide), and when adherence is high. The most important benefit of any health measure taken, however, would be a significant reduction in the mortality rate, as caused by Covid-19.

In the present study, we therefore examined, across eight countries whether higher adherence to behavioral NPIs predicted a smaller increase in Covid-19 mortality over a period of three months in a prospective longitudinal study with quasi-experimental design. Building on and extending earlier work on the link between macrosocial factors and mental health [15, 16], we selected the United States, Russia, Poland, Sweden, Germany, France, Spain and the United Kingdom for our study. These countries not only represent different types of societies and health care systems, but also differ in their emphasis on personal freedom, government effectiveness and attitudes to NPIs [15–18]. In most countries, the first cases of Covid-19 were reported in January 2020 (except Poland, which reported first cases in March 2020) and governments and health authorities subsequently advocated behavioral measures to contain the pandemic. With the exception of Sweden, all countries declared total or partial lockdowns in March 2020, which were eased from April or May 2020 (see Table 1). Table 1 shows the times of lockdowns in spring 2020 and the governmental NPIs in the eight investigated countries that were effective in the end of May 2020 and in the beginning of June 2020.

**Table 1. Times of lockdowns in spring 2020 and nonpharmaceutical interventions (NPIs) in the eight investigated countries between the end of May 2020 and the beginning of June 2020.**

| | Russia | Poland | Sweden | USA | Germany | France | Spain | UK |
|---|---|---|---|---|---|---|---|---|
| **Begin of lockdown** | March | March | - | March | March | March | March | March |
| **First easing of lockdown** | May | April | - | April | April/May | May | April/May | May |
| **"Stay-at-home" order (whole country or single states/provinces)** | X | X | - | X | X | X | X | X |
| **Compulsory wearing of face masks** | X | X | - | X | X | X | X | X |
| **Social distancing (1.5 to 2 meters / 4.9 to 6.6 feet)** | X | X | - | X | X | X | X | X |

Source of information are the country specific governmental sites [19–26].

## Methods

### Procedure and participants

The overall sample was comprised of 7,658 participants from eight countries: United States: $N$ = 904, Russia: $N$ = 986, Poland: $N$ = 924, Sweden: $N$ = 922, Germany: $N$ = 917, France: $N$ = 940, Spain: $N$ = 960, and United Kingdom: $N$ = 1,105. Demographics of all samples are presented in Table 2. Data were collected within ten days between May 28 and June 7, 2020 by an independent social marketing and research institute (YouGov, www.yougov.de) through online population-based panel surveys in the national language of the countries. The participants were recruited from the resident population, and were aged 18 years and older. To achieve representativeness, a stratification by age, gender and region was performed. In all countries, participation was compensated by panel-specific tokens that can be converted into vouchers or cash payments. The main research work took place in Germany. The study was approved by the ethics committee of the Faculty of Psychology of the Ruhr-Universität Bochum (Germany) and pre-registered with AsPredicted.org on May 25, 2020 (https://aspredicted.org/e7a9g.pdf). All required permits and approvals for the data collection in the eight countries were obtained by the independent social marketing and research institute You-Gov. All participants were properly instructed and gave their informed consent to participate online. The dataset used in the present study is available in S1 Dataset.

**Table 2. Demographic variables (total and individual samples).**

|  | All | Russia | Poland | Sweden | USA | Germany | France | Spain | UK |
|---|---|---|---|---|---|---|---|---|---|
| **N with valid data** | 7,658 | 986 | 924 | 922 | 904 | 917 | 940 | 960 | 1,105 |
| **Gender (female, %)** | 53 | 54.7 | 54.7 | 51 | 51.8 | 51 | 57.7 | 51 | 52.2 |
| **Age groups (%)** |  |  |  |  |  |  |  |  |  |
| **18 to 24 years** | 8.1 | 7.8 | 9.6 | 6.8 | 8.7 | 6.7 | 8.5 | 6.3 | 10 |
| **25 to 34 years** | 16.6 | 20.9 | 17.5 | 20.9 | 13.5 | 12.8 | 14.8 | 14 | 17.9 |
| **35 to 44 years** | 16.5 | 20.3 | 18.9 | 8.8 | 15.6 | 14.4 | 14.9 | 21.3 | 16.9 |
| **45 to 54 years** | 18.4 | 17.3 | 15.4 | 19 | 18.6 | 19.8 | 18.5 | 20.9 | 18 |
| **55 years and older** | 40.4 | 33.7 | 38.5 | 44.5 | 43.6 | 46.3 | 43.3 | 37.6 | 37.1 |
| **Marital Status (%)** |  |  |  |  |  |  |  |  |  |
| **Single** | 23.4 | 16.5 | 20.5 | 32.9 | 22 | 23.8 | 21.8 | 23.3 | 26.3 |
| **Romantic relationship, not married** | 16.4 | 11.6 | 17.3 | 22.5 | 7.6 | 14.9 | 22.8 | 18.5 | 16 |
| **Married** | 47.8 | 57.4 | 50.4 | 35.8 | 55.5 | 45.8 | 43.6 | 48.4 | 45.7 |
| **Widowed, divorced** | 12.4 | 14.5 | 11.8 | 8.9 | 14.8 | 15.5 | 11.8 | 9.7 | 11.9 |
| **Social Status (%)** |  |  |  |  |  |  |  |  |  |
| **Lower class** | 5.1 | 2.8 | 3.8 | 5.1 | 7.6 | 7.6 | 7.6 | 4.1 | 2.7 |
| **Working class** | 22.2 | 19.1 | 15.8 | 20.7 | 16.4 | 18.4 | 19.3 | 31.4 | 33.8 |
| **Lower middle class** | 25.9 | 37.3 | 32.6 | 13.6 | 19.1 | 25.3 | 26.7 | 21 | 30.2 |
| **Middle middle class** | 36.8 | 36.8 | 36.3 | 46.6 | 39.6 | 38.9 | 32.8 | 36.9 | 28.4 |
| **Upper middle class** | 9 | 3.4 | 8.9 | 12.8 | 15.9 | 9.2 | 12 | 6.6 | 4.9 |
| **Upper class** | 1 | 0.5 | 2.7 | 1.2 | 1.3 | 0.5 | 1.7 | 0.1 | - |
| **Living Environment (%)** |  |  |  |  |  |  |  |  |  |
| **Large city** | 42.3 | 77.3 | 48.8 | 47.7 | 38.4 | 35.1 | 28.9 | 37.9 | 25.5 |
| **Small city** | 35 | 19.6 | 36.6 | 33 | 39.2 | 36 | 39.7 | 41.7 | 35.4 |
| **Rural community** | 22.7 | 3.1 | 14.6 | 19.3 | 22.5 | 28.9 | 31.4 | 20.4 | 39.1 |

Due to rounding, the sum of the frequencies is not always 100%.

## Measures

**Adherence to governmental anti-Covid-19 measures.** Adherence was measured using the question "How much do you adhere to the rules to combat the Corona crisis?" on a 5-point Likert scale (0 = *not at all*, 1 = *little*, 2 = *moderate*, 3 = *strong*, 4 = *very strong*).

**Mortality.** Mortality data for June 1, 2020 and August 31, 2020 were taken from published sources that receive data from the Covid-19 Data Repository by the Center for Systems Science and Engineering (CSSE) at Johns Hopkins University (JHU, USA) [27]. Fig 1 provides the full course of reported Covid-19 mortality from February 17 to August 31, 2020 in the eight countries studied.

## Statistical analyses

Statistical analyses were conducted using SPSS 24. After descriptive analyses, the relationship between self-reported adherence to the behavioral anti-Covid-19 measures and the percentage increase in mortality from June 1, 2020 to August 31, 2020 was assessed via a zero-order bivariate correlation analysis at the level of countries ($N$ = 8) and tested for significance ($p <$ .05, two-tailed).

## Results

Of the total of 7,658 participants, 73.3% stated that their adherence to the behavioral measures was strong or very strong (see Fig 2). Lower than average proportions of strong or very strong adherence were reported by participants in the USA, Sweden, Poland and Russia (48.6% = lowest), higher than average in Germany, France, Spain and the United Kingdom (88.0% = highest). Mean adherence ratings varied from 2.48 (Russia) to 3.35 (United Kingdom). The highest variability within countries was found in the United States ($SD$ = 1.14), the lowest in the United Kingdom ($SD$ = 0.80) (see Table 3). Remarkably, the level of self-reported adherence in the different countries predicted the relative increase in country-wide Covid-19 deaths over the next three months.

In the countries studied, a total of 216,613 deaths due to Covid-19 was reported by June 1, 2020 [27]. By August 31, 2020, the total number of deaths had increased by 47% to a new total of 318,430 [27]. However, as shown in Fig 2, there were large differences between countries in the mortality increase. In absolute terms, the increase ranged from 687 in Germany to 78,686 in the USA. The smallest percentage increase was recorded in France (+6.3%) and the largest in Russia (+264.2%). It is striking that the percentage increase in mortality in the four countries with poorer adherence was almost ten times higher (81.3%) than in the four countries with better adherence (8.4%). In absolute numbers, the low adherence countries had 93,288 additional Covid-19 deaths between June 1 and August 31, while the high adherence countries had "only" 8,529 additional deaths. The correlation between the percentage of the population reporting strong or very strong adherence and the percentage increase in Covid-19 mortality of the countries was $r$ = -0.91 (Pearson product-moment correlation, $N$ = 8, $p <$ .002), which expresses a strong effect ($R^2$ = 0.83).

## Discussion

The Covid-19 pandemic has significantly changed the daily lives of many people around the world in recent months. As SARS-CoV-2 spreads from person to person in the community, wearing face masks and maintaining social distance has become part of everyday life in many countries [6, 18]. The success of such measures, however, depends largely on the willingness of the population to adhere to them, in addition to their actual utility when used properly. Our

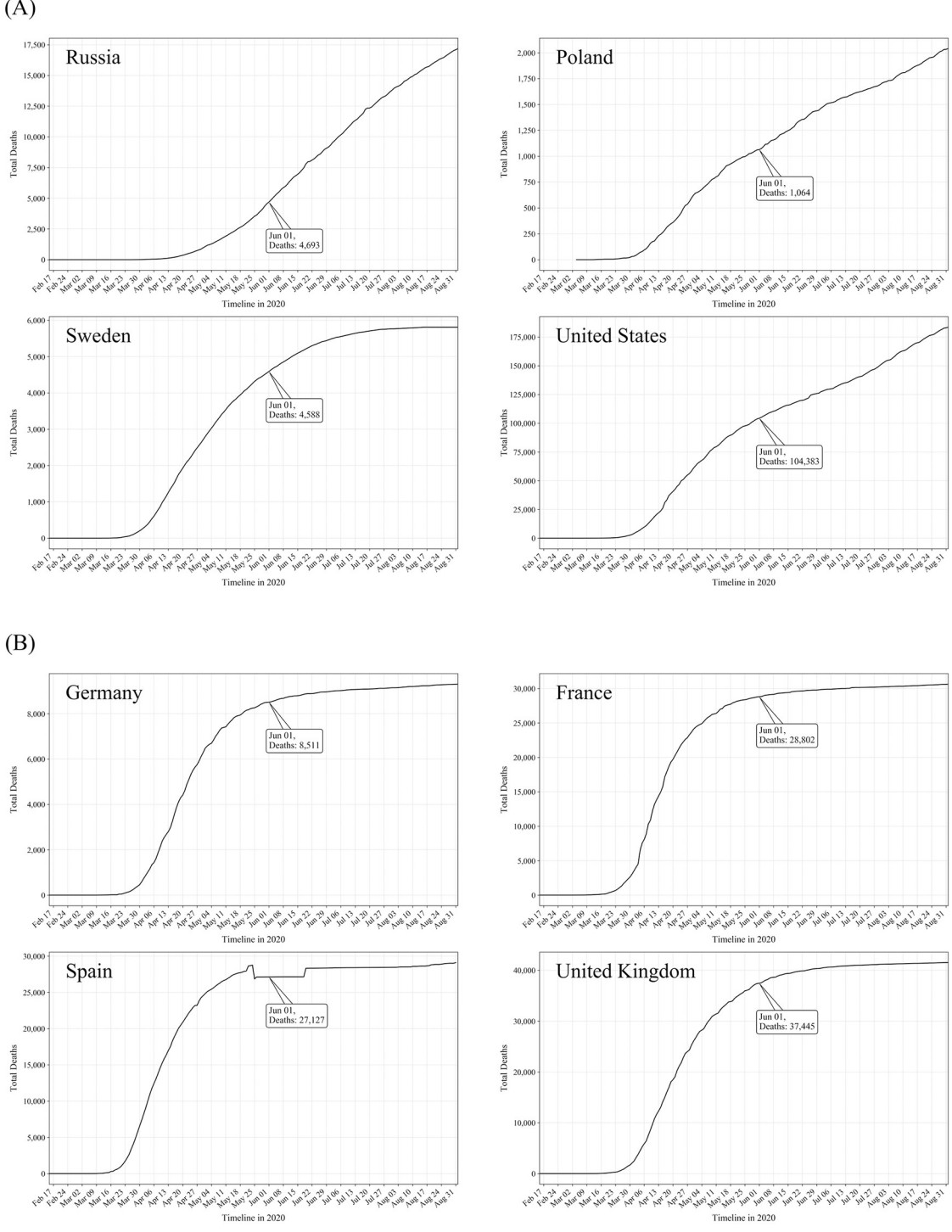

**Fig 1. Course of reported Covid-19 mortality from February 17 to August 31, 2020 in the eight countries studied.** The number of deaths on June 1, 2020 is indicated in the boxes for each country: (A) Countries with low self-reported adherence as of June 1, 2020; (B) Countries with high self-reported adherence as of June 1, 2020 (see [27]). Notes. The decrease, plateau and increase of the mortality rate in Spain through the course of June is due to inconclusive official data reports in this time period.

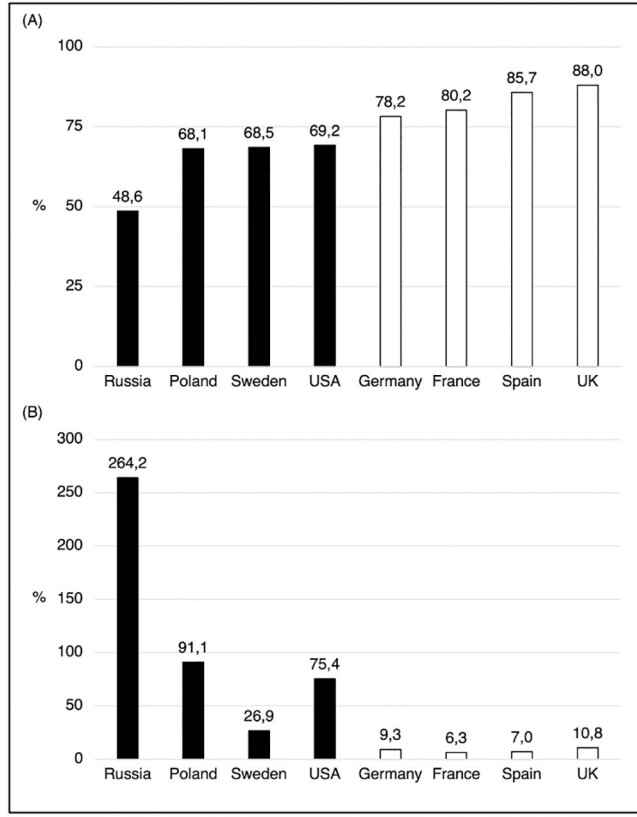

**Fig 2. Adherence to behavioral NPIs to mitigate Covid-19 and subsequent mortality increases over three months in eight countries: (A) Percentage of adult population with strong or very strong self-reported adherence on June 1, 2020; (B) Percentage increase in reported Covid-19 mortality from June 1 to August 31, 2020.** Notes. Dark bars: countries with lower adherence, light bars: countries with higher adherence (median split).

present results show that adherence to protective behavioral measures is followed by significantly lower mortality from Covid-19 at the community level. The considerable differences in adherence as reported around June 1, 2020, were strongly predictive of the increase in Covid-19 mortality over the next three months. The lowest adherence to the Covid-19 mitigation behaviors was in the two post-Soviet transition countries Russia and Poland followed by the USA and Sweden, where governments and authorities demonstrated an ambivalent to sometimes dismissive attitude towards the measures. The highest adherence was reported from countries who had previously suffered very high mortality rates (United Kingdom, Spain, France) or whose population showed a very positive perception of government communication (Germany).

In addition to the form and perception of the governmental communication, further factors can impact the individual adherence to the anti-Covid-19 measures. For instance, many people use social media such as Facebook and Twitter as source of Covid-19 information [28]. In contrast to other information sources such as television reports, newspaper reports and official sites of federal government and authorities, content provided on social media is user-generated. Thus, each user can create, modify and share the content [29, 30]. As a consequence, social media often provide a high amount of unfiltered (mis)information [31]. Previous research on earlier extraordinary societal situations (e.g., terrorist attacks, epidemics) [32, 33] and recent research on Covid-19 [34, 35] showed that the consumption of such information

**Table 3. Self-reported adherence to behavioral anti-Covid-19 measures on June 1st, 2020 and published Covid-19 mortality in 8 participating countries on June 1 and August 31, 2020.**

| Country | Self-reported adherence to Covid-19 mitigation behaviors | | Covid-19 deaths as of | | Increase in Covid-19 deaths from June-August 2020 | |
|---|---|---|---|---|---|---|
| | % of the population with strong or very strong adherence | Mean adherence rating on 0–4 scale (mean +/- sd) | June 1st, 2020 (*N*) | August 31st, 2020 (N) | Absolute (*N*) | Relative (%) |
| 1. Russia | 48.6 | 2.48 +/- 1.03 | 4,693 | 17,093 | 12,400 | 264.22 |
| 2. Poland | 68.1 | 2.79 +/- 1.07 | 1,064 | 2,033 | 969 | 91.1 |
| 3. Sweden | 68.5 | 2.85 +/- 0.94 | 4,588 | 5,821 | 1,233 | 26.9 |
| 4. Unites Sates | 69.2 | 2.88 +/- 1.14 | 104,383 | 183,069 | 78,686 | 75.4 |
| 5. Germany | 78.2 | 3.02 +/- 0.92 | 8,511 | 9,298 | 787 | 9.3 |
| 6. France | 80.2 | 3.07 +/- 0.94 | 28,802 | 30,606 | 1,804 | 6.3 |
| 7. Spain | 85.7 | 3.29 +/- 0.83 | 27,127 | 29,011 | 1,884 | 6.9 |
| 8. United Kingdom | 88.0 | 3.35 +/- 0.80 | 37,445 | 41,499 | 4,054 | 10.8 |
| All countries | 73.3 | 2.97 +/- 1.00 | 216,613 | 318,430 | 101,817 | 47.0 |
| Low adherence countries (#1–4) | 63.6 | 2.75 +/- 1.05 | 114,728 | 208016 | 93,288 | 81.3 |
| High adherence countries (#5–8) | 83.0 | 3.18 +/- 0.87 | 101,885 | 110,414 | 8,529 | 8.4 |

Results are shown are the percentage of participants with strong or very strong adherence, means and standard deviations of adherence ratings, the number of Covid-19 deaths on June 1 and August 31, 2020, as well as the increase in the number of deaths between the two dates (absolute numbers and percentages) for each country separately, all countries and countries with low or high adherence ratings (median split).

can contribute to emotional overload, enhanced stress symptoms, experience of burden and reduction of adherence to urgent measures. To prevent the negative impact of social media use and to increase adherence to the NPIs, a stronger control of the content provided on social platforms by the providers is urgent. In addition, governmental communication should stress the responsibility for the generated online content of each user and the need to verify all information through official sources before sharing.

Moreover, individual cost-benefit considerations can influence the level of adherence to the NPIs. Measures such as social distancing, limited leisure travel, wearing of face masks, frequent washing of hands and fever measuring reduce the risk for infection and thus in the longer-term contribute to the reduction of the pandemic spread and of the mortality rate [11]. However, in the short-term, they can be experienced as inconvenient and restrictive [2]. People who perceive the short-term costs of the measures as higher and more significant than the longer-term benefits tend to low adherence to the NPIs, especially when they rate their own risk for infection as low [13, 36]. Therefore, programs that focus specifically on these individuals and emphasize the longer-term benefits of adherence are required.

A further factor that might impact the adherence to the NPIs is sense of control. Sense of control belongs to important humans needs [37]. People with a low level of sense of control often have enhanced stress and anxiety symptoms. They tend to rumination and maladaptive coping-strategies such as problematic substance use [38]. In a recent study, low sense of control was positively associated with the experience of burden by the Covid-19 situation [39]. Against this background, it can be assumed that people with a low level of sense of control are at risk for low adherence to the NPIs because they do not trust in the efficacy of their own activities and thus are convinced that their behavior cannot contribute to the pandemic fight. Governmental communication should emphasize that the adherence to the anti-Covid-19

measures of each individual is important for the control of the current situation and thus for the pandemic fight.

There are some limitations to the present study. First, it must be noted that due to the very dynamic circumstances, the present results represent a snapshot of the Covid-19 situation in the summer of 2020 in the eight countries studied. Second, apart from the Asian part of Russia, no Asian, African or South American countries were included in the present study, which limits the generalizability of the findings. Third, the present study is quasi-experimental, drawing inferences from self-reported adherence assessed at only one measurement time point to country-wide mortality figures, and thus may be subject to participant biases, and can only illuminate correlational relationships, and not causation. Furthermore, adherence was assessed by a single-item measure. Available cross-sectional and longitudinal research reported single-item scales that measure various psychological and behavioral constructs such as risk-taking to have adequate psychometric properties [40–44]. Nevertheless, future studies are recommended to include measures that assess adherence to specific NPIs at several measurement time points to gain a more detailed view of the Covid-19 situation. Fourth, data of adherence were collected by an independent social marketing and research institute. To achieve representativeness, stratification by age, gender and region was performed. Thereby, the age distribution was quoted and weighted representative of the population from 18 years of age in each country. Thus, only respondents aged 18 and older were considered; respondents under 18 were not taken into account in the age distribution. As a consequence, higher proportions of the samples fall to the older population groups. Finally, the reported figures on Covid-19 mortality in different countries are likely to be significantly influenced by different data collection methods among hospitals and government agencies, as well as testing frequencies, and differing levels of unreported cases. These data collection methods were not under the control of the present study.

Despite limitations of the present study, the nearly tenfold difference in additional mortality found here between countries with low and high adherence is so strong that study limitations as a sole explanation for these differences seems unlikely. Furthermore, the present study is a true prospective prediction over a period of three months. It therefore seems reasonable to assume that higher adherence is indeed associated with reduced mortality. A major short-term challenge for societies and governments, therefore, is to foster the highest possible levels of adherence to anti-Covid-19 NPIs. The cost of life-saving measures can vary greatly, often reaching tens of thousands to millions of dollars per year saved [45]. Hand washing has long been one of the most cost-effective interventions. In the current pandemic, wearing a mask and keeping your distance could now be added to this age-old, cost-effective life-saving measure. With the exception of Russia, at least two thirds of the population in the countries studied indicate strong or very strong adherence to behavioral measures to mitigate Covid-19. The present data lend support to such adherence, and may be referred by individuals and campaigns as they seek to justify adoption and continued use of NPIs and to find the appropriate to balance between the interests of individual freedom and economic well-being with health and survival in a situation that has been termed "the perfect moral storm" [10].

## Supporting information

**S1 Dataset. Dataset used for analyses in present study.**
(SAV)

## Acknowledgments

We thank Dr. Kristen Lavallee for proof-reading the manuscript.

## Author Contributions

**Conceptualization:** Jürgen Margraf, Julia Brailovskaia, Silvia Schneider.

**Data curation:** Jürgen Margraf, Silvia Schneider.

**Formal analysis:** Jürgen Margraf.

**Investigation:** Jürgen Margraf, Silvia Schneider.

**Methodology:** Jürgen Margraf, Julia Brailovskaia, Silvia Schneider.

**Project administration:** Jürgen Margraf, Silvia Schneider.

**Resources:** Jürgen Margraf.

**Validation:** Jürgen Margraf.

**Visualization:** Jürgen Margraf, Julia Brailovskaia.

**Writing – original draft:** Jürgen Margraf, Silvia Schneider.

**Writing – review & editing:** Jürgen Margraf, Julia Brailovskaia, Silvia Schneider.

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
