## [Decision Letter · Decision Letter 0]

26 Jan 2021

PONE-D-20-36481

Adherence to behavioral Covid-19 mitigation measures strongly predicts mortality

PLOS ONE

Dear Dr. Margraf,

Thank you for submitting your manuscript to PLOS ONE. After careful consideration, we feel that it has merit but does not fully meet PLOS ONE’s publication criteria as it currently stands. Therefore, we invite you to submit a revised version of the manuscript that addresses the points raised during the review process.

Please consider the reviewers comments and respond. 

We look forward to receiving your revised manuscript.

Kind regards,

Andrew Soundy

Academic Editor

PLOS ONE

Journal Requirements:

2.) In your Methods section, please provide additional information about the participant recruitment method and the demographic details of your participants. Please ensure you have provided sufficient details to replicate the analyses such as: a) the recruitment date range (month and year), b) a description of how participants were recruited, and c) descriptions of where participants were recruited and where the research took place.

Moreover, please provide a rationale for the inclusion of the specific 8 countries, and for investigating mortality rates between in the period  June - August 2020. "

3.)  We note your statement "All required permits and approvals for the data collection in the eight countries were obtained". Please specify the names of the IRBs that approved the study in the 8 countries.

4.) We noted in your submission details that a portion of your manuscript may have been presented or published elsewhere.

"The present data set has been used for the preparation of an other manuscript that is submitted to PLOS ONE (PONE-D-20-27521R1), but that paper has another main focus. Editor for the other manuscript is Antonio Scala."

Please clarify whether this publication was peer-reviewed and formally published. If this work was previously peer-reviewed and published, in the cover letter please provide the reason that this work does not constitute dual publication and should be included in the current manuscript.

Reviewers' comments:

Reviewer's Responses to Questions

**Comments to the Author**

1. Is the manuscript technically sound, and do the data support the conclusions?

Reviewer #1: Yes

2. Has the statistical analysis been performed appropriately and rigorously? 

Reviewer #1: Yes

3. Have the authors made all data underlying the findings in their manuscript fully available?

Reviewer #1: Yes

4. Is the manuscript presented in an intelligible fashion and written in standard English?

Reviewer #1: Yes

5. Review Comments to the Author

Reviewer #1: Abstract

Keeping distance should read maintaining social or interpersonal distance

Benefits questioned by whom?

Syntax and grammatical mistakes evident.

Introduction

Page 1- Line 56-Rather than dynamics this should state reduce ‘R’ associated with transmission

Page 1- Line 59- this should be re-written as it suggests the introduction was written in May of 2021 when most countries are currently in lockdown. Authors should state that in May of 2020 there was some easing of government implemented ant-pandemic measures

Syntax and grammatical mistakes evident

Method & Results

Is there an explanation for the far higher percentage of individuals aged over 45 across all countries. Perhaps a table would be useful describing the specific anti-pandemic measures in place at the time of data collection across the participant countries. This is important as the study rests on the idea that adherence is related to morality and therefore the paper might be stronger if the specifics for each country are clearly identified.

Its of critical importance that the number of deaths from the virus in each of the participating countries is accurate. Therefore, the source of this information should be clearly identified. Do the figures match on WHO statistics? There is a reference (19) provided but what is the validity and accuracy of this data?

There is no breakdown of adherence relative to age. This might be useful to consider.

Discussion

There really should be a richer discussion of the psychological factors involved in non-adherence other than whether governments were ambivalent about them.

Accuracy of single item to assess adherence and single time point measure should be addressed

6. PLOS authors have the option to publish the peer review history of their article (what does this mean?). If published, this will include your full peer review and any attached files.

Reviewer #1: **Yes: **Ian M Grey

---

## [Author Response · Author response to Decision Letter 0]

12 Feb 2021

Editor:

Thank you for submitting your manuscript to PLOS ONE. After careful consideration, we feel that it has merit but does not fully meet PLOS ONE’s publication criteria as it currently stands. Therefore, we invite you to submit a revised version of the manuscript that addresses the points raised during the review process.

Response to Editor:

Thank you very much for the possibility to submit a revised version of our manuscript. We have considerably revised the original manuscript by addressing each point raised by the reviewer and the additional requirements of the Journal. This remarkably improved our work. We really hope to have covered all the suggestions adequately.

1.) Please ensure that your manuscript meets PLOS ONE's style requirements, including those for file naming. …

Response to 1.:

Our manuscript meets PLOS ONE’s style requirements, including those for file naming.

2.) In your Methods section, please provide additional information about the participant recruitment method and the demographic details of your participants. Please ensure you have provided sufficient details to replicate the analyses such as: a) the recruitment date range (month and year), b) a description of how participants were recruited, and c) descriptions of where participants were recruited and where the research took place.

Moreover, please provide a rationale for the inclusion of the specific 8 countries, and for investigating mortality rates between in the period June - August 2020. "

Response to 2.:

The information you requested is included in the manuscript. Table 2 presents the demographic details of the samples. The information about the participant recruitment method is included in the Method section. We have provided sufficient details to replicate the analyses:

a) The recruitment was conducted within ten days between May 28 and June 7, 2020. 

b) and c) Participants were recruited by an independent social marketing and research institute (YouGov, www.yougov.de) through an online population-based panel. We added the URL of the online-page of YouGov in the revised manuscript to enable readers to get more information about this independent social marketing and research institute.

The authors of the present study are located in Germany (see the affiliations). Thus, the work on the research data was conducted in Germany. We included this information in the revised manuscript.

Our research group has previously conducted two cross-national studies on these eight countries due to their different welfare system (please see for detailed explanation Scholten, Velten, & Margraf, 2018; Scholten, Velten, Neher, & Margraf, 2017). Against this background, the present data collection has been considered as a continuation of our previous research. Moreover, the eight countries not only represent different types of societies and health care systems, but also differ in their emphasis on personal freedom, government effectiveness and attitudes to behavioral anti-Covid-19 measures (e.g., Czeisler et al., 2020; Liang et al., 2020). We included this information in the Introduction section of the revised manuscript. Furthermore, following the advice of the reviewer, we included a table with country specific anti-Covid-19 measures in the revised manuscript. 

Data on adherence in the eight countries were collected between May 28 and June 7, 2020. Considering that in June 2020 nobody knew how the Covid-19 situation will develop, we did not a priori plan to assess further data after the data collection. However, considering the worldwide high mortality rate caused by Covid-19 in the summer months, it was urgent to understand its predictors to contribute to the pandemic fight. Therefore, we decided posteriori to investigate whether adherence assessed in June can predict the mortality rate. Considering our significant findings, a timeframe of three months is informative for this issue. We hope that you consider this explanation as sufficient.

References:

Czeisler, M. É., Tynan, M. A., Howard, M. E., Honeycutt, S., Fulmer, E. B., Kidder, D. P., . . . Baldwin, G. (2020). Public attitudes, behaviors, and beliefs related to COVID-19, stay-at-home orders, nonessential business closures, and public health guidance—United States, New York City, and Los Angeles, May 5–12, 2020. Morbidity and Mortality Weekly Report, 69(24), 751. doi:10.15585/mmwr.mm6924e1

Liang, L.-L., Tseng, C.-H., Ho, H. J., & Wu, C.-Y. (2020). Covid-19 mortality is negatively associated with test number and government effectiveness. Scientific reports, 10(1), 1-7. doi:10.1038/s41598-020-68862-x

Scholten, S., Velten, J., & Margraf, J. (2018). Mental distress and perceived wealth, justice and freedom across eight countries: The invisible power of the macrosystem. PLoS One, 13(5), e0194642. 

doi:10.1371/journal.pone.0194642

Scholten, S., Velten, J., Neher, T., & Margraf, J. (2017). Wealth, justice and freedom: objective and 

subjective measures predicting poor mental health in a study across eight countries. SSM-Population 

Health, 3, 639-648. doi:10.1016/j.ssmph.2017.07.010

3.) We note your statement "All required permits and approvals for the data collection in the eight countries were obtained". Please specify the names of the IRBs that approved the study in the 8 countries.

Response to 3.:

The study was approved by the ethics committee of the Faculty of Psychology of the Ruhr-Universität Bochum. In addition, the independent social marketing and research institute YouGov that conducted the data collection obtained all required permissions and approvals for the conduction of international data collections. We included this information in the revised manuscript.

4.) We noted in your submission details that a portion of your manuscript may have been presented or published elsewhere.

"The present data set has been used for the preparation of an other manuscript that is submitted to PLOS ONE (PONE-D-20-27521R1), but that paper has another main focus. Editor for the other manuscript is Antonio Scala."

Please clarify whether this publication was peer-reviewed and formally published. If this work was previously peer-reviewed and published, in the cover letter please provide the reason that this work does not constitute dual publication and should be included in the current manuscript.

Response to 4.:

The publication was peer-reviewed and formally published: 

Margraf, J., Brailovskaia, J., & Schneider, S. (2020). Behavioral measures to fight COVID-19: An 8-

country study of perceived usefulness, adherence and their predictors. PLoS One, 15(12), e0243523. 

doi:10.1371/journal.pone.0243523

We included the reason that this work does not constitute dual publication and should not be included in the current manuscript in the cover letter.

Reviewer:

To Reviewer:

Thank you very much for the insightful comments. We have considerably revised the original manuscript by addressing each point raised by you. This remarkably improved our work. We really hope to have covered all the suggestions adequately.

1. Abstract: Keeping distance should read maintaining social or interpersonal distance

Benefits questioned by whom?

Response to 1.:

We apologize the unclear formulation. We reformulated both sentences in the revised manuscript.

2. Syntax and grammatical mistakes evident.

Response to 2.:

Thank you for this advice. The revised manuscript has been proof-read by an English native speaker.

3. Introduction

Page 1- Line 56-Rather than dynamics this should state reduce ‘R’ associated with transmission

Response to 3.: 

We corrected this formulation in the revised manuscript.

4. Page 1- Line 59- this should be re-written as it suggests the introduction was written in May of 2021 when most countries are currently in lockdown. Authors should state that in May of 2020 there was some easing of government implemented ant-pandemic measures

Response to 4.:

We corrected this formulation in the revised manuscript.

5. Syntax and grammatical mistakes evident

Response to 5.:

Thank you for this advice. The revised manuscript has been proof-read by an English native speaker.

6. Method & Results

Is there an explanation for the far higher percentage of individuals aged over 45 across all countries. 

Perhaps a table would be useful describing the specific anti-pandemic measures in place at the time of data collection across the participant countries. This is important as the study rests on the idea that adherence is related to morality and therefore the paper might be stronger if the specifics for each country are clearly identified.

Response to 6.:

Participants were recruited by an independent social marketing and research institute (YouGov, www.yougov.de) through an online population-based panel. To achieve representativeness, a stratification by age, gender and region was performed. Following the stratification rules of YouGov, the age distribution is quoted and weighted representative of the population from 18 years of age in each of the eight countries. Only respondents aged 18 and older are considered; respondents under 18 years are accordingly not taken into account in the age distribution. As a consequence, higher proportions fall to the older population groups. This explains the higher percentages of individuals aged over 45 across the investigated countries. We included this information to the limitations in the revised manuscript.

Following your suggestion, we included a table describing the specific anti-pandemic measures at the time of data collection across the participant countries in the revised manuscript.

7. Its of critical importance that the number of deaths from the virus in each of the participating countries is accurate. Therefore, the source of this information should be clearly identified. Do the figures match on WHO statistics? There is a reference (19) provided but what is the validity and accuracy of this data?

Response to 7.:

We agree that it is of critical importance that the number of deaths from the virus of each of the participating countries is accurate. The number of deaths from the virus in each of the participating 

countries is taken from Hasell et al. (2020). This is an online-page that provides daily updated Covid-19 data from each country that we investigated. The data come from the Covid-19 Data Repository by the Center for Systems Science and Engineering (CSSE) at Johns Hopkins University (JHU). Thus, the data on the number of deaths are accurate. We included this information in the revised manuscript.

Reference:

Hasell, J., Mathieu, E., Beltekian, D., Macdonald, B., Giattino, C., Ortiz-Ospina, E., . . . Ritchie, H. (2020). A cross-country database of COVID-19 testing. Scientific data, 7(345), 1-7. doi:10.1038/s41597-020-00688-8

8. There is no breakdown of adherence relative to age. This might be useful to consider.

Response to 8.:

Considering your concern, we calculated the relationship between the variables age group and adherence for the overall sample (r = .087, p < .001) and for each of the eight investigated countries 

separately (GE: r = .224, p < .001, FR: r = .069, p < .05, ES: r = .070, p < .01, PL: r = .118, p < .001, RU: r = -.001, n.s., SV: r = .080, p < .05, UK: r = .065, p < .01, US: r = .071, p < .05). In most of the investigated countries, this relationship is positive, but weak. This is an interesting issue, but its inclusion in the main manuscript would interrupt the reading flow because the relationship between age and adherence is not the main topic of the present investigation. We hope that you understand this. If you consider this as important, we can include the correlation results as supplementary material.

9. Discussion

There really should be a richer discussion of the psychological factors involved in non-adherence other than whether governments were ambivalent about them.

Response to 9.:

Following your advice, we included a richer discussion of the psychological factors that could be involved in non-adherence in the revised manuscript.

10. Accuracy of single item to assess adherence and single time point measure should be addressed

Response to 10.:

Following your advice, we addressed the accuracy of a single item measure to assess adherence and of a single time point measure in the revised manuscript.

---

## [Decision Letter · Decision Letter 1]

18 Mar 2021

Adherence to behavioral Covid-19 mitigation measures strongly predicts mortality

PONE-D-20-36481R1

Dear Prof Margraf,

We’re pleased to inform you that your manuscript has been judged scientifically suitable for publication and will be formally accepted for publication once it meets all outstanding technical requirements.

Kind regards,

Andrew Soundy

Academic Editor

PLOS ONE

Additional Editor Comments (optional):

Reviewers' comments:

Reviewer's Responses to Questions

**Comments to the Author**

1. If the authors have adequately addressed your comments raised in a previous round of review and you feel that this manuscript is now acceptable for publication, you may indicate that here to bypass the “Comments to the Author” section, enter your conflict of interest statement in the “Confidential to Editor” section, and submit your "Accept" recommendation.

Reviewer #1: All comments have been addressed

2. Is the manuscript technically sound, and do the data support the conclusions?

Reviewer #1: Yes

3. Has the statistical analysis been performed appropriately and rigorously? 

Reviewer #1: Yes

4. Have the authors made all data underlying the findings in their manuscript fully available?

Reviewer #1: Yes

5. Is the manuscript presented in an intelligible fashion and written in standard English?

Reviewer #1: Yes

6. Review Comments to the Author

Reviewer #1: Thank you for responding to my comments. I am satisfied with the changes that have been made to the manuscript. I recommend publication.

7. PLOS authors have the option to publish the peer review history of their article (what does this mean?). If published, this will include your full peer review and any attached files.

Reviewer #1: No

---

## [Editor Report · Acceptance letter]

19 Mar 2021

PONE-D-20-36481R1 

Adherence to behavioral Covid-19 mitigation Measures strongly predicts Mortality 

Dear Dr. Margraf:

I'm pleased to inform you that your manuscript has been deemed suitable for publication in PLOS ONE. Congratulations! Your manuscript is now with our production department. 

Kind regards, 

on behalf of

Dr. Andrew Soundy 

Academic Editor

PLOS ONE